# Structure-preserving visualization for single-cell RNA-Seq profiles using deep manifold transformation with batch-correction

Yongjie Xu [1,2], Zelin Zang [1,2], Jun Xia [1,2], Cheng Tan[1,2], Yulan Geng[2] & Stan Z. Li [2 ✉]

Dimensionality reduction and visualization play an important role in biological data analysis, such as data interpretation of single-cell RNA sequences (scRNA-seq). It is desired to have a visualization method that can not only be applicable to various application scenarios, including cell clustering and trajectory inference, but also satisfy a variety of technical requirements, especially the ability to preserve inherent structure of data and handle with batch effects. However, no existing methods can accommodate these requirements in a unified framework. In this paper, we propose a general visualization method, deep visualization (DV), that possesses the ability to preserve inherent structure of data and handle batch effects and is applicable to a variety of datasets from different application domains and dataset scales. The method embeds a given dataset into a 2- or 3-dimensional visualization space, with either a Euclidean or hyperbolic metric depending on a specified task type with type *static* (at a time point) or *dynamic* (at a sequence of time points) scRNA-seq data, respectively. Specifically, DV learns a structure graph to describe the relationships between data samples, transforms the data into visualization space while preserving the geometric structure of the data and correcting batch effects in an end-to-end manner. The experimental results on nine datasets in complex tissue from human patients or animal development demonstrate the competitiveness of DV in discovering complex cellular relations, uncovering temporal trajectories, and addressing complex batch factors. We also provide a preliminary attempt to pre-train a DV model for visualization of new incoming data.

[1] Zhejiang University, Hangzhou 310058, China. [2] AI Division, School of Engineering, Westlake University, Hangzhou 310024, China.
✉email: stan.zq.li@westlake.edu.cn

The advent of technologies that interrogate genome-scale molecular information at single-cell resolution, such as single-cell RNA sequencing and mass cytometry, provides important insight into the comprehensive analysis of cellular differentiation and the relationship between cells[1]. Although single-cell RNA sequences (scRNA-seq) data have high dimensionality, their intrinsic dimensionality is typically low because many genes are co-expressed and the droplet-based scRNA-seq is very sparse (> 90% genes with zero counts in a typical cell profile). Therefore, dimensionality reduction and visualization methods play an important role in interpreting scRNA-seq datasets, such as extracting effective information, intuitively understanding data distribution, and interpreting the relationship between cells[2,3].

In this paper, we address the following application scenarios: Firstly, we develop a machine learning method with the ability to preserve geometric structure of the high dimensional scRNA-seq data in the dimensionality reduced space and visualization of scRNA-seq data that can be applied to both cell clustering and trajectory inference tasks. These two scenarios are closely related yet have different technical goals: (1) For cell clustering is to explore the relationship between different cell types at a given time[4–17], which we call the *static* (at a time point) scenario. It is to learn a low-dimensional embedding in which cells belonging to the same type should be close to each other whereas those of different types be away from each other. (2) For trajectory inference, or the *dynamic* (at a sequence of time points) scenario, the learning is to uncover temporal trajectories of cells[17–21], which characterizes the transition process of immature cells into mature cells with specific types. Secondly, the still another task we are addressing is (3) performing batch correction to build low-dimensional representations of the biological contents of cells disentangled from the technical variations[17]. Thirdly, based on the above tasks, we make another two preliminary attempts including (4) building a "batch invariant" model to embed new incoming data impacted by diverse factors[17] and (5) building a pre-trained model to embed new incoming heterogeneous data. In contrast, the current methods are inflexible and generally follow a uniform assumption when facing with different application scenarios and unable to accommodate these requirements in a unified framework.

Traditional linear/nonlinear dimensionality reduction methods have grown explosively during the last decades, including local geometric structure preservation and global geometric structure preservation methods. The former aims to find a subspace by preserving the local geometric structure such as locally linear embedding (LLE)[4], Laplacian eigenmaps (LE)[5] and stochastic neighbor embedding (SNE)[6]. The latter tries to preserve the global characteristics of input data in a low dimensional subspace, such as principal component analysis (PCA)[7], isometric mapping (ISOMAP)[18], diffusion map (DM)[19] and PHATE[20]. These methods are often insufficient to mine underlying biological information as they consider global or local structure preservation alone. Furthermore, t-distributed stochastic neighbor embedding (t-SNE)[8] and uniform manifold approximation and projection (UMAP)[9] based on manifold learning have demonstrated excellent performance in capturing complex local and global geometric structures of biological data. However, both of them suffer from several limitations. Firstly, t-SNE is not robust in the presence of technical noise and tends to form spurious clusters from randomly distributed data points, producing misleading results that may hinder biological interpretation[16]. Meanwhile, t-SNE preserves the local clustering structures, but the global structures such as inter-cluster relationships and distances cannot be reliably preserved. Secondly, the addition of new data points to existing embeddings is infeasible due to the non-parametric nature of t-SNE and UMAP. Instead, they need to be rerun on the combined dataset,

which is computationally expensive and is not scalable. Thirdly, the "cell-crowding" problem (e.g., t-SNE), "cell-mixing" (e.g., UMAP) problem, and the lack of batch correction capability will affect the effectiveness of data visualization when handling large-scale datasets with hundreds of thousands of cells[22].

In recent years, deep neural networks (DNNs)[23] have been utilized as effective non-linear dimensionality reduction and visualization tools for processing large datasets, incorporating different factors, and improving the scalable ability of models. This field mainly involves two mainstream directions, including (1) Deep manifold learning methods, such as parametric UMAP[10], Markov-Lipschitz deep learning (MLDL)[11], deep manifold transformation (DMT)[12], deep local-flatness manifold embedding (DLME)[24], EVNet[13], unified dimensional reduction neural-network (UDRN)[14] and IVIS[15], and (2) Deep reconstruction learning methods, which covers various (variational) autoencoders[16,25,26]. Generally speaking, the latter seeks to reconstruct the input data distribution and often ignores the importance of intrinsic geometric structure in input data. In contrast, The former preserves the geometric structure of raw data as much as possible, which is beneficial for mining the underlying information of biological data but lacks batch correction ability. Specifically, these methods usually suffer three fundamental issues: (1) High distortion embedding problem. Most methods assume the embedding space is Euclidean, which is not enough for modeling and analyzing *dynamic* scRNA-seq data because the Euclidean geometry is not optimal for representing the hierarchical and branched developmental trajectories. As shown in Bourgain's theorem, Euclidean space is unable to obtain comparably low distortion for tree data, even when using an unbounded number of dimensions[27]. For example, Poincaré maps (Poin_maps)[21] was proposed recently to harness the power of hyperbolic geometry into the realm of *dynamic* scRNA-seq data analysis. (2) Deep manifold learning methods do not have the ability to preserve the geometric structure of the high dimensional scRNA-seq data and correct batch effects in an end-to-end manner. Most methods require multiple separate steps, each with its own method, including batch correction (e.g., Seurat3 CCA[2], Harmony[28], LIGER[29], fastMNN[30], Scanorama[31], SAUCIE[32], scVI[26] and Conos[33]), dimensionality reduction, and visualization. Recently, Ding et al.[17] proposed a scalable deep generative model scPhere based on variational autoencoders to embed cells into a low-dimensional hyperspherical or hyperbolic space to better capture the inherent properties of scRNA-seq data. ScPhere addresses multi-level, complex batch factors, facilitates the interactive visualization of large datasets, resolves "cell-crowding" problem, and uncovers temporal trajectories. (3) Poor flexibility for various application scenarios. Most methods tend to follow a uniform assumption for different application scenarios (e.g., *static* or *dynamic* data with or without batch effects) without considering the inherent characteristics of biological data, and the pre-trained reference model lacks suitable preprocessing steps to accommodate new incoming heterogeneous data and can only map new incoming homogeneous data.

To address the above challenges, we propose a general visualization model, deep visualization (DV), that preserves the inherent structure of scRNA-seq data and handles complex batch effects. Specifically, DV learns a structure graph based on local scale contraction to describe the relationships between cells more accurately, transforms the data into 2- or 3-dimensional embedding space while preserving the geometric structure of the data, and constructs a priori batch effect graph to correct batch effects in an end-to-end manner. For *static* scRNA-seq data, we minimize the structure distortion between structure graph and visualization graph in Euclidean space (DV_Eu). For *dynamic* data, to better represent and infer underlying hierarchical and

branched developmental trajectories, we embed cells to the hyperbolic space with Poincaré (DV_Poin) or Lorentz (DV_Lor) model and visualize the embeddings in a Poincaré disk. We demonstrate the superior performance of DV on critical existing cases based on nine diverse datasets from human, mouse, and model organisms, including processing large *static* and *dynamic* scRNA-seq datasets with/without complex multilevel batch effects, visualizing cell profiles from highly complex tissues and developmental processes. We also make a preliminary attempt to build a pre-trained reference model for visualization of new incoming homogeneous and heterogeneous data. Overall, our method serves as a unified solution for enhanced representation, complex batch correction, visualization, and an interpretation tool for single-cell genomics research.

## Results

For the purpose of *static* and *dynamic* scRNA-seq data visualization, DV embeds the data into a 2- or 3-dimensional Euclidean or hyperbolic latent space at the end of the DNNs (Fig. 1a), in terms of the curvature characteristics of the data manifold. A Euclidean space with zero curvature is commonly adopted by most visualization methods (e.g., t-SNE, UMAP, PHATE, and IVIS) for its flatness and intuitive class boundaries, which may be sufficient for exploring the relationship between different cell types in *static* data. Hyperbolic embedding with negative curvature has been proposed for learning of latent representations from hierarchical textual and graph-structured data[27]. We believe that it is suitable for *dynamic* data to uncover temporal trajectories. This is because in such type of data, the exponential growth of the number of leaves in a tree with respect to its depth is analogous to the exponential growth of surface area with respect to its radius. In the hyperbolic space, circle circumference and disc area grow exponentially with radius, as opposed to the Euclidean space where they only grow linearly and quadratically[34].

DV model assumes that a good embedding should preserve the geometric structure of scRNA-seq data as much as possible. According to *manifold assumption*, the observed data are low dimensional manifolds uniformly sampled in a high dimensional Euclidean space. In practice, the droplet-based scRNA-seq data usually has a large number of zero or near-zero values. The relationship between cells is difficult to be defined by vector similarity (e.g., Euclidean distance) directly for such high-dimensional sparse data. Therefore, DV learns a reliable structure graph based on local scale contraction between each cell and its corresponding augmented cells (linear interpolation between each cell and its $k$ neighbor points) to describe the relationship between cells more accurately. Specifically, DV estimates the underlying manifold structure in four main steps (Fig. 1a): Firstly, constructing a fully connected undirected structure graph $G_{structure}$ for cells and their corresponding augmentation cells based on structure embedding learned by the structure module, where each node corresponds to an individual cell and each edge has a weight (Euclidean distance between the structure embeddings of the two connected cells). The purpose is to estimate the local geometry of the underlying topological manifold. Secondly, DV learns a low-dimensional Euclidean or hyperbolic embedding per cell and constructs a fully connected undirected visualization graph $G_{visualization}$ for cells and their corresponding augmentation cells based on visualization embedding learned by the visualization module. In detail, for Euclidean latent space, DV learns 2-dimensional embeddings and adopts Euclidean distance to describe the relationship between embeddings. For hyperbolic latent space, DV learns 2 or 3-dimensional embeddings with Poincaré or Lorentz model and adopts hyperbolic distance to describe the relationship between embeddings. Thirdly, DV

converts $G_{structure}$ and $G_{visualization}$ edge weight from distance to similarity based on the student's t-distribution. The purpose is to highlight similar pairwise nodes and weaken dissimilar pairwise nodes. These steps are commonly used in manifold learning (e.g., t-SNE, UMAP) to approximate the structure of an unknown manifold from similarities in the feature space. Finally, to preserve the geometric structure of scRNA-seq data, DV adopts the geometric structure preservation loss function to train DNNs, which minimizes the distribution discrepancy between $G_{structure}$ and $G_{visualization}$. To make DV compatible with the batch correction ability simultaneously, we integrate the manually designed priori batch effect graph $G_{batch}$ into the $G_{visualization}$ to be learned in the training process to learn a $G_{visualization}$ with batch effect removed. As a deep learning model trained by mini-batch stochastic gradient descent, DV is especially suited to process large scRNA-seq datasets with complex multilevel batch effects and facilitates emerging applications. We provide complete details in the "Methods" section.

**DV preserves the structure of scRNA-seq data in very low-dimensional spaces in visualizing large datasets.** Applying DV to scRNA-seq data, we systematically assess the visualization performance of DV embeddings in a latent space with few (2 or 3) dimensions. We compare the geometric structure preservation performance ($Q_{global}$ and $Q_{local}$ scores, interpreted as a scRNA-seq dataset comprises a smooth manifold and a good dimensionality reduction method would preserve local and global distances on this manifold) of DV, which embeds cells in Euclidean or hyperbolic spaces, as well as to PCA[7], t-SNE[8], UMAP[9], IVIS[15], PHATE[20], Poin_maps[21] and hyperbolic scPhere (scPhere_wn)[17]. Following scPhere[17], we apply DV to seven scRNA-seq datasets from human and mouse, spanning from small (thousands) to very large number (hundreds of thousands) of cells from one or multiple tissues, and with a small (two) to very large (dozens) of expected cell types. The "small" datasets are: (1) a blood cell dataset[35] with only 10 erythroid cell profiles and 2293 CD14+ monocytes, (2) 3314 human lung cells[36], (3) 1378 mouse white adipose tissue stromal cells[37], and (4) 1755 human splenic nature killer cells spanning four subtypes[38]. The "large" datasets are: (1) 35,699 retinal ganglion cells (RGC) in 45 cell subsets[39], (2) 599,926 cells spanning 102 subsets across 59 human tissues in the human cell landscape (HCL)[40], and (3) 86,024 C. elegans embryonic cells (ELEGAN) collected along a time course from <100 min to >650 min after each embryo's first cleavage[41].

DV obtains more competitive performance compared with baseline methods on "small" datasets (Supplementary Fig. 1 and Fig. 2). It is worth noting that DV_Poin and DV_Lor with hyperbolic latent spaces even perform better than DV_Eu in some datasets, although there are discrete cell types in these datasets. Overall, these methods achieve good visualizations for these smaller datasets without batch effects but with minor challenges. For example, (1) In human lung cells, all methods mix pericyte and APC except DV_Lor (Supplementary Fig. 1m). (2) PHATE (Supplementary Figs. 2f, p) and Poin_maps (Supplementary Figs. 1i, s) – are designed for development trajectories – connecting cells inaccurately when only discrete cell types are present. (3) PCA cannot effectively capture complex data structures due to the lack of nonlinear ability, resulting in the distortion issue with the increased number of cell types (Supplementary Fig. 2d). (4) Compared with the dimensionality reduction methods based on manifold learning (e.g., DV, UMAP, and t-SNE), scPhere_wn (Supplementary Figs. 2j, t) based on variational auto-encoder does not have enough power to aggregate similar cells belonging to the same type and push away dissimilar cells belonging to different types.

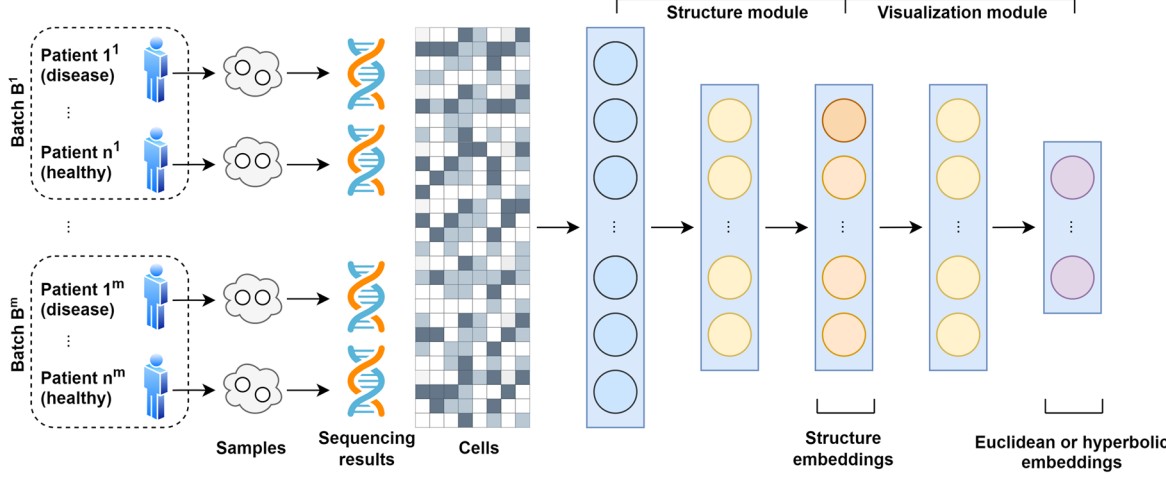

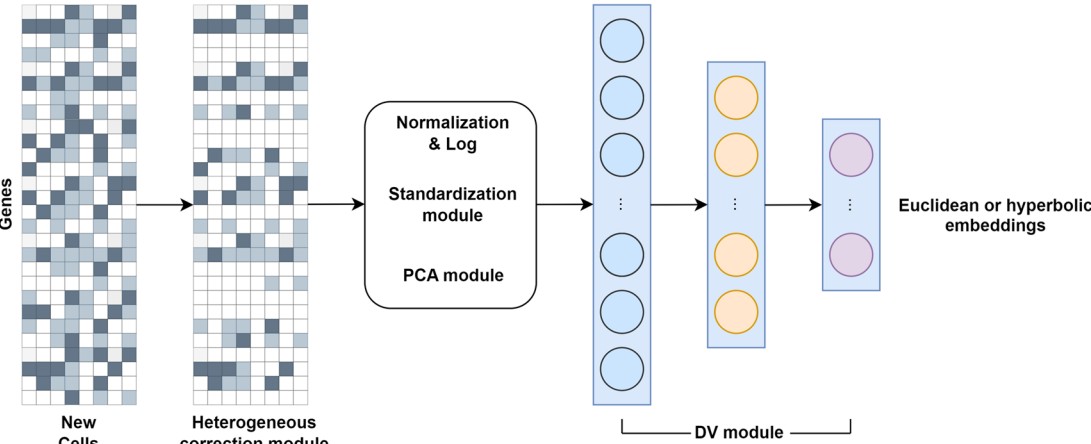

**Fig. 1 The Deep Visualization (DV) model. a** The DV framework. DV takes as input scRNA-seq measurements of multilevel technical or biological factors (e.g., replicate patient, disease) and learns the latent structure of cells while taking into consideration of batch effect. **b** DV learns a structure graph from the input based on local scale contraction, then in the process of preserving the geometric structure of scRNA-seq data, disentangles semantic visualization graph with batch effect into semantic visualization graph with batch-effect removed and priori batch effect graph. **c** The preprocessing modules for heterogeneous new datasets and the learned DV model are used for mapping new datasets to the reference.

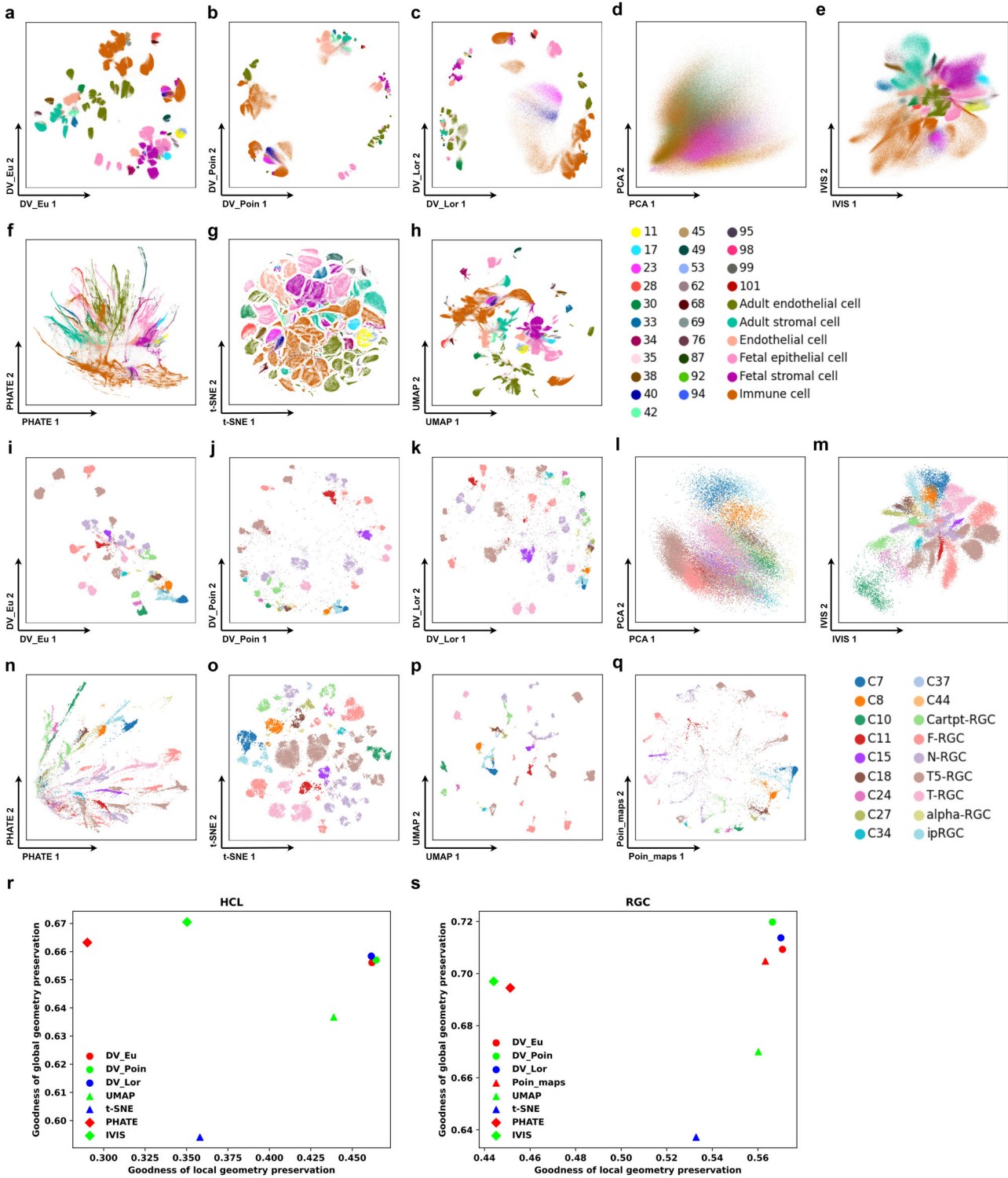

**Fig. 2 DV preserves local and global structures in visualizing large *static* scRNA-seq data. a–q** DV learns latent representations that provide excellent visualization of local and global structure, even in very large datasets. DV_Eu learns representations in the Euclidean space (**a**, **i**). DV_Poin learns representations in the hyperbolic space with Poincaré model (**b**, **j**). DV_Lor learns representations in the hyperbolic space with Lorentz model (**c**, **k**). 2-dimensional PCA (**d**, **l**), IVIS (**e**, **m**), PHATE (**f**, **n**), t-SNE (**g**, **o**), UMAP (**h**, **p**) and Poin_maps (**q**) representations for human cell landscape (HCL) (GSE134355) (**a–h**) and mouse retinal ganglion cells (RGC) (GSE137400) (**i–q**) with cells colored by major cell types. Quantify global/hierarchical and local structure preservation performance of HCL and RGC datasets (**r**, **s**).

DV obtains predominant advantages over baseline methods on datasets with a larger number of cells and clusters, such as mouse RGC cells and human HCL cells. As shown in Fig. 2, although t-SNE, UMAP and Poin_maps can distinguish individual cell types among RGC (Fig. 2a–h, Supplementary Figs. 3a–h) and HCL (Fig. 2i–q, Supplementary Fig. 3i–q) well, DV_Eu, DV_Poin and DV_Lor achieve superior local and global structure preservation performance (Fig. 2r, s, Supplementary Data 1). In contrast,

DV_Poin and DV_Lor based on hyperbolic latent spaces can obtain better global structure preservation performance than DV_Eu based on Euclidean latent space in RGC cells, especially DV_Poin based on the Poincaré ball model. Specifically, Cartpt-RGC clusters are close together in DV_Poin hyperbolic latent space (Fig. 2j), while in other methods except Poin_maps (Fig. 2q), they are embedded in different locations of the latent space, this means that hyperbolic latent space can better preserve the hierarchical global structure of cells. In HCL cells, there are six major cell groups, including fetal stromal cells, fetal epithelial cells, adult endothelial cells, endothelial cells, adult stromal cells and immune cells, and their respective clusters are close to each other in DV_Eu (Fig. 2a), but are more dispersed in t-SNE embeddings (Fig. 2g). HCL cells contain two cell sources, including adult cells and fetal cells, their respective clusters are close to each other and have better differentiation between both sources in DV_Eu (Fig. 2a), but are more mixed in UMAP embeddings (Fig. 2h). Compared with DV_Eu, DV_Poin (Fig. 2b) and DV_Lor (Fig. 2c) have larger volume for structure storage, thus they gain more power to push away dissimilar clusters, this will help with data analysis in scenarios without priori labels/celltypes, such as visualization partition analysis (e.g., DV_Poin hyperbolic visualization space contains six major regions). Furthermore, some clusters (e.g., enterocyte cells have an immune function) belonging to adult endothelial cells are close to immune cells (mostly B cells). Meanwhile, UMAP and t-SNE are often unable to effectively visualize datasets with a larger number of cells as reflected both visually. For example, the "cell-crowding" problem existing in t-SNE (Fig. 2g), different clusters are uniform spread across the visualization space, leading to the poor ability of t-SNE to recognize the major clusters and mine the distinct cell types, and the "cell-mixing" problem existing in UMAP (Fig. 2h), different clusters are twisted and mixed. DV overcomes the above issues, because it learns a more reliable structure graph $G_{structure}$ based on nonlinear DNNs and is trained using mini-batches, while t-SNE and UMAP are learned using all the data, and their hyperparameters (e.g., "perplexity" in t-SNE) have to be adapted to larger number of cells, but increasing the "perplexity" makes t-SNE computationally expensive simultaneously (Supplementary Fig. 16). DV is natural to process a large number of cells with a time complexity that is linear with the number of input cells. As expected, PCA (Fig. 2d, l), IVIS (Fig. 2e, m) and PHATE (Fig. 2f, n) do not perform well for these large datasets with mostly discrete cell types.

DV obtains more intuitive visualization on *dynamic* cells, which are expected to show developmental trajectories, such as from stem cells to mature cells (Fig. 3j–r), and explains the relationships between clusters (Fig. 3a–i). As described above, DV_Poin (Fig. 3b) and DV_Lor (Fig. 3c) have larger volumes for hierarchical structure storage, thus the dataset can be divided into multiple regions to facilitate the analysis of cell differentiation for each major cell type individually. For example, there are four major regions (muscle cells, excretory cells, pharyngeal cells, neuron cells and other individual clusters) in DV_Poin latent space if not considering NA cells. Compared with DV, data in the central region of baseline methods are faced with the severe "cell-crowding" problem (Fig. 3m–r), this is not conducive to a more accurate recognition of cell origin. However, the problem is well solved by DV, where the data in the central region can be clearly identified and assigned to different differentiation branches (Fig. 3j–l). Moreover, we can position the expected root cells of the developmental process at the center of a Poincaré disk, then the distance of each cell from the center can be thought of as a pseudo time. For a specific cell type, we can see cells progress with distance continuously in the Poincaré disk at almost a fixed angle. Fortunately, compared with scPhere_wn (Fig. 3r), DV (Fig. 3j–l)

can automatically place root cells near the center of the Poincaré disk without prior knowledge of the root label, which is convenient for data analysis when unknown time labels. Furthermore, DV_Poin (Fig. 3b) and DV_Lor (Fig. 3c) alleviate the "cell-mixing" problem (e.g., seam cells and hypodermis cells are entangled) existing in DV_Eu (Fig. 3a) and UMAP (Fig. 3h). Therefore, DV embeds *dynamic* cells into a hyperbolic space with Lorentz or Poincaré model, which is suitable for differentiated data analysis, and optionally converts the coordinates in the Lorentz model to the Poincaré disk for 2-dimensional visualization. More importantly, DV retains the biological explanation brought by the scPhere_wn method. Specifically, in ELEGAN cells, the cells are ordered neatly in the latent space by both time and lineage, from a clearly discernible root at time 100–130 at the center of the Poincaré disk (cells from < 100 were mostly unfertilized germline cells) to cells from time > 650 near the border of the Poincaré disk or away from the origin in the Poincaré and Lorentz model (Fig. 3k, l, Supplementary Fig. 13). Within the same cell type, cells are ordered by embryo time in the Poincaré disk (Fig. 3b, c). After first appearing along a developmental trajectory, cells of the same type progress with embryo time, forming a continuous trajectory occupying a range of angles[17]. Moreover, different cell types (e.g., ciliated amphid neurons, ciliated nonamphid neurons, hypodermis, seam cells and body wall muscle) that appear at slightly different embryonic time points, have their origins around the same region and progress with embryonic time in a similar way, forming a continuous trajectory but at a different angle and/or distance ranges from the center[17]. These patterns are harder to discern in IVIS (Fig. 3n), PHATE (Fig. 3o), t-SNE (Fig. 3p) and UMAP (Fig. 3q), where cells from consecutive time points are compacted, cells that appear early are relatively distant from each other in the embeddings, and temporal progression is not in the same direction. Thus, the DV model with a hyperbolic latent space learns smooth (in time) and interpretable cell trajectories.

**DV effectively models complex, multilevel batch, and other variables.** In realistic biological datasets, scRNA-seq profiles are typically impacted by diverse factors, including technical batch effects in separate experiments and different lab protocols, as well as biological factors, such as inter-individual variation, sex, disease or tissue location. However, most batch-correction methods can handle only one batch variable and may not be well-suited to the increasing complexity of current datasets. Applying DV to scRNA-seq data with multiple known confounding factors (e.g., batches and conditions), we systematically assess the performance of DV embeddings in a latent space with few (2, 3) or low (5, 10, 20) dimensions by comparing the geometric structure preservation performance ($Q_{global}$ and $Q_{local}$ scores, interpreted as a complex multi-batch scRNA-seq dataset comprises multiple smooth manifold and a good dimensionality reduction method will preserve local and global distances on each manifold after removing batch effect) and classification performance ($ACC_{mvo}$ score, interpreted as a good batch correction method will integrate different manifolds) of DV, which embeds cells in Euclidean space for *static* cells and hyperbolic space for *dynamic* cells, as well as to Euclidean scPhere (scPhere_normal), hyperspherical scPhere (scPhere_vmf), scPhere_wn and other visualization methods, including t-SNE, UMAP, IVIS and PHATE (with 50 principal components, batch-corrected by Harmony or scVI). Following scPhere[17], we apply DV to a dataset of 301,749 cells profiled in a complex experimental design from the colon mucosa of 18 patients with ulcerative colitis (UC), a major type of inflammatory bowel disease (IBD), and 12 healthy individuals[42]. The *static* datasets are: (1) 26,678 stromal cells and glia (12 cell

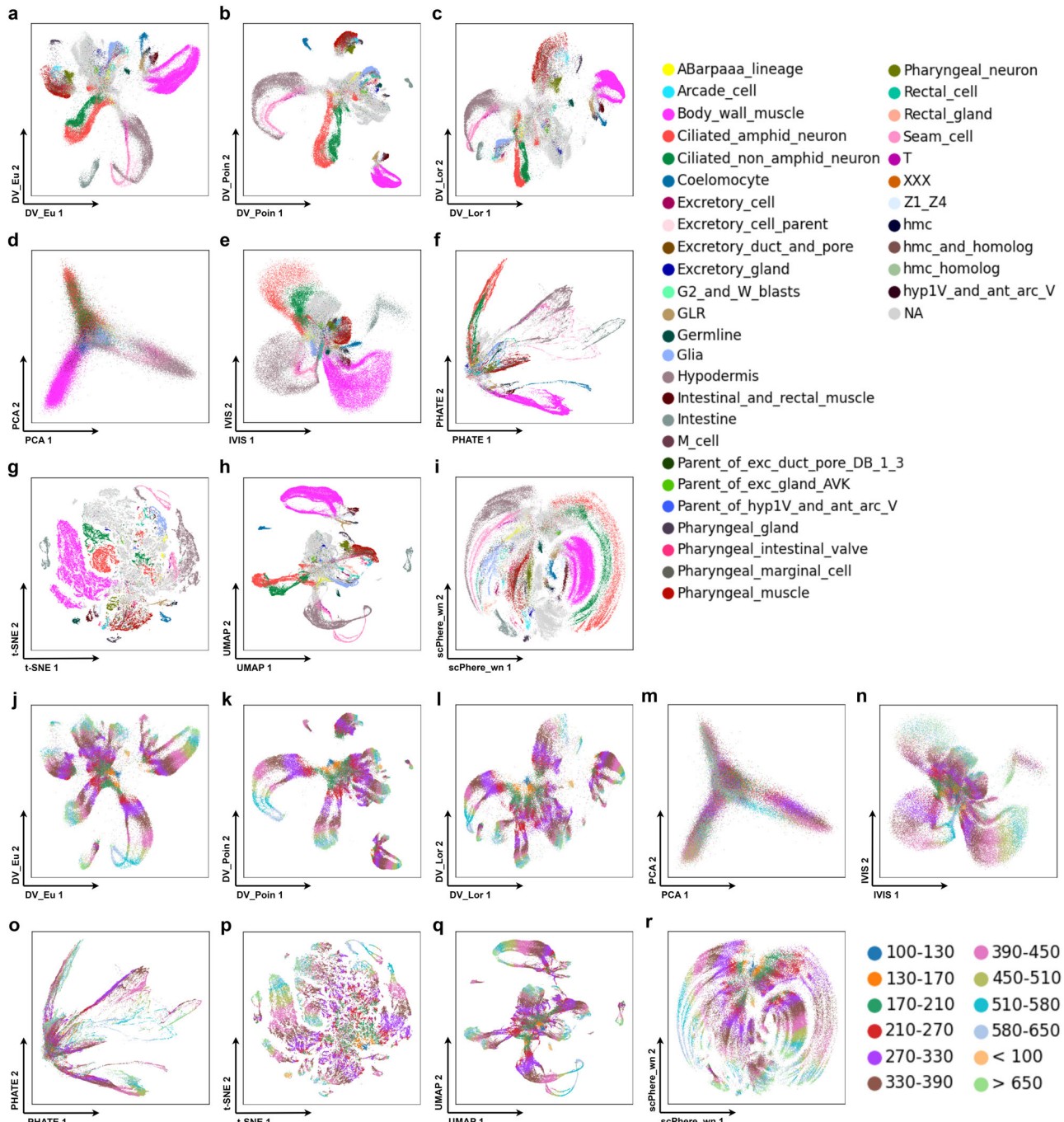

**Fig. 3 DV shows developmental trajectories in visualizing large scRNA-seq *dynamic* data.** DV_Eu learns representations in the Euclidean space (**a**, **j**). DV_Poin learns representations in the hyperbolic space with Poincaré model (**b**, **k**). DV_Lor learns representations in the hyperbolic space with Lorentz model (**c**, **l**). 2-dimensional PCA (**d**, **m**), IVIS (**e**, **n**), PHATE (**f**, **o**), t-SNE (**g**, **p**), UMAP (**h**, **q**) and scPhere_wn (**i**, **r**) representations for ELEGAN (GSE126954) with cells colored by cell types (**a**–**i**) and time phases (**j**–**r**).

types), and (2) 210,614 immune cells (23 cell types). The *dynamic* dataset are: (1) 64,457 epithelial cells (12 cell types), and (2) 86,024 ELEGAN cells (12 time states).

DV_Eu obtains more competitive performance compared with baseline methods on *static* datasets with a larger number of cells and multiple confounding factors, such as stromal dataset (30 patients with patient origin and disease status factors) and immune dataset (30 patients with patient origin, disease status and location factors). Analyzing cells with the patient origin and disease status (healthy, uninflamed and inflamed) as the batch vector, not only recapitulates the main cell groups, but also allows

us to better visually explore cellular relations (Supplementary Fig. 4a–i) and finds cell groups related to disease directly. For example, in the stromal dataset, the postcapillary venules cells, endothelial cells and microvascular cells are close to each other, and adjacent to the pericyte (Supplementary Fig. 4a). Conversely, these distinctions can barely be discerned in a UMAP (Supplementary Fig. 4f) and Poin_maps (Supplementary Fig. 4g) plot of the same data, where endothelial and microvascular cells are very close. Among fibroblasts, cells are arranged in a manner that mirrors their position along the crypt-villus axis, from RSPO3+ cells, to WNT2B+ cells, to WNT5B+ cells. Strikingly,

the inflammatory fibroblasts, which are unique to UC patients and are independent of the patient origin, are readily visible (Supplementary Fig. 4a, pale green) and are both distinct from the other fibroblasts, while spanning the range of the "crypt-villus axis"[17]. Considering the ability to integrate disease status, DV_Eu merges part inflammatory fibroblasts with WNT2B+ fibroblasts (Supplementary Fig. 4a). When learning a DV_Eu model that includes patient origin, disease status, and anatomical region as the batch vectors, immune cells groups visually by cell type (Fig. 4a), and the influence of patient origin, disease status and region is largely removed. For example, the CD8+IL17+ T cells are nestled between CD8+ T cells and activated CD4+ T cells in a manner that was intriguing and consistent with the mixed features of those cells (Fig. 4a)[17]. In terms of evaluation criteria, on the one hand, DV_Eu (batch correction on 2-dimensional embeddings) obtains competitive local ($Q_{local}$ score) and global ($Q_{global}$ score) geometric structure preservation performance compared with UMAP, IVIS and PHATE based on Harmony or scVI (batch correction on 50 principal components) and outperforms scPhere_normal and t-SNE on the stromal dataset (Supplementary Fig. 4j, k) and immune dataset (Fig. 4i, j, Supplementary Data 2). Moreover, DV_Eu achieves better performance when batch correction on low (5, 10 and 20) dimensional embeddings. On the other hand, based on 5-nearest neighbors (5-NN) classification accuracy of cell types, DV_Eu obtains competitive batch correction visualization performance ($ACC_{mvo}$ score on 2-dimensional embeddings) compared with IVIS, PHATE, t-SNE, UMAP and Poin_maps combined with Harmony or scVI on the stromal dataset (Supplementary Fig. 4l) and immune dataset (Fig. 4k, Supplementary Data 3). Moreover, DV_Eu obtains a significant batch correction advantage ($ACC_{mvo}$ score on 5, 10 and 20-dimensional embeddings) over scPhere method on low dimensional embeddings (Fig. 4l, Supplementary Fig. 4m, Supplementary Data 3). It is worth noting that while some methods outperform DV on the $Q_{local}$, $Q_{global}$ or $ACC_{mvo}$ score on 2-dimensional embeddings, they are not always stable. For example, IVIS and PHATE combined Harmony obtains good $Q_{local}$ and $Q_{global}$ scores on the stromal dataset, while their $ACC_{mvo}$ scores are very poor.

DV_Poin and DV_Lor obtain more intuitive visualization and competitive performance compared with baseline methods on *dynamic* datasets, such as epithelial dataset (30 patients with patient origin, disease status and location factors) and ELEGAN dataset (7 batches). In epithelial cells, we readily discern developmental ordering from intestinal stem cells to terminally differentiated cells in the Poincaré disk (Fig. 5b, c), with stem cells at the center of the disk for intuitive interpretation: one trajectory is from stem cells to secretory TA cells, to immature goblet cells, to goblet cells, the other trajectory is from stem cells to TA2 cells, to immature enterocyte cells, to enterocyte cells. In contrast, developmental trajectories are less apparent when we embed cells in Euclidean space of scPhere_normal (Fig. 5f). The 2-dimensional visualization embeddings of t-SNE (Fig. 5g), UMAP (Fig. 5h), DV_Eu (Fig. 5a), IVIS (Fig. 5d), PHATE (Fig. 5e), scPhere_wn (Fig. 5j) and scPhere_vmf (Fig. 5i) are reasonable, although the t-SNE has some small spurious clusters, goblet cells have one spurious cluster close to enterocytes in DV_Eu, IVIS, scPhere_wn and scPhere_vmf, several cell types (M-cells and TA2 cells, tuft and enteroendocrine cells) are merged in PHATE, and the developmental trajectories are less apparent when the cell types are missing in DV_Eu, scPhere_vmf and scPhere_wn. In addition, DV_Poin and DV_Lor obtain a significant local (Fig. 5n, Supplementary Data 2) and global (Fig. 5o, Supplementary Data 2) geometric structure preservation advantage compared with other methods in Euclidean space, and this advantage increases significantly when the dimension of

embedding space is increased. In ELEGAN cells, the cellular relations and developmental trajectories in DV and scPhere_wn have minor changes when compared Supplementary Fig. 5 (with batch correction) with Fig. 3 (without batch correction), which indicates that the batch effect has little influence on this dataset, but the former still mitigate the less apparent batch effect problem existing in some cell types (e.g., abarpaaa lineage, ciliated non amphid neurons and ciliated amphid neurons). Moreover, the advantages of DV_Poin and DV_Lor (Supplementary Fig. 14) in this dataset described in the previous section (Supplementary Fig. 13) still remain when compared with t-SNE, UMAP and scPhere (Supplementary Fig. 15).

DV_Eu achieves impressive results on all stromal, epithelial, and immune cells simultaneously (Supplementary Fig. 6a), demonstrating its capacity to embed large numbers of cells of diverse types, states and proportions. The 2-dimensional visualization embeddings of t-SNE (Supplementary Fig. 6b) and UMAP (Supplementary Fig. 6c) using Harmony batch-corrected results accounting for the patient status as inputs are reasonable. However, when removing the PCA preprocessing, t-SNE (Supplementary Fig. 6g) and UMAP (Supplementary Fig. 6h) fail to an unsuccessful visualization, while DV_Eu (Supplementary Fig. 6f) can achieve competitive performance compared with scPhere_wn (Supplementary Fig. 6e) and scPhere_vmf (Supplementary Fig. 6i). For example, there are a lot of noise points existing in t-SNE embeddings, which illustrates that t-SNE can not distinguish different cell types, and UMAP suffers from a severe confusion issue among different clusters. Moreover, this also reflects that the Harmony method can not effectively remove the batch effect problem in sparse data. For example, the plasma cells are separated in UMAP embeddings. Overall, these results demonstrate the superior performance of DV_Eu compared to the combination of Harmony's batch correction and t-SNE or UMAP's visualization through multiple experiments on large datasets with a large number of cells and cell types, multilevel batch effects, and complex structures.

**Pre-trained reference DV builds atlases for visualization and annotation of new incoming data.** As a parametric model, we can train DV to co-embed new incoming/testing data to a latent space learned from training data only. We use DV to map cells from new incoming patients, one critical homogeneous application case as multiple studies need to be integrated by training a "batch-invariant" DV model, the other critical heterogeneous application case as a new study can be explored by a pre-trained DV model. Then, DV takes the gene expression vectors or PCA principal components of cells as inputs and maps them to a 2-dimensional Euclidean latent space to achieve data visualization and annotation. Therefore, we conduct preliminary experiments to explore the scalable ability of DV to train general models based on large-scale datasets.

For homogeneous case (training data and testing data share the same gene names and numbers), we learn a "batch-invariant" DV_Eu model for stromal, epithelial and immune cells from 18 patients training data of the UC dataset and use it to visualize the cells from 12 patients testing data directly. Then we train k-nearest neighbor (k-NN) classifiers (k = 5) on 2-dimensional embeddings of the training data and apply the learned k-NN classifiers to 2-dimensional embeddings of the testing data. The experiment results demonstrate that DV's embeddings of testing data with high quality. For the stromal dataset (Supplementary Fig. 7), DV_Eu (72.84%) obtains better classification accuracy compared with scPhere_normal (71.25%), scPhere_wn (70.25%) and scPhere_vmf (70.24%), it significantly improves the precision of most categories, especially on inflammatory fibroblasts cells,

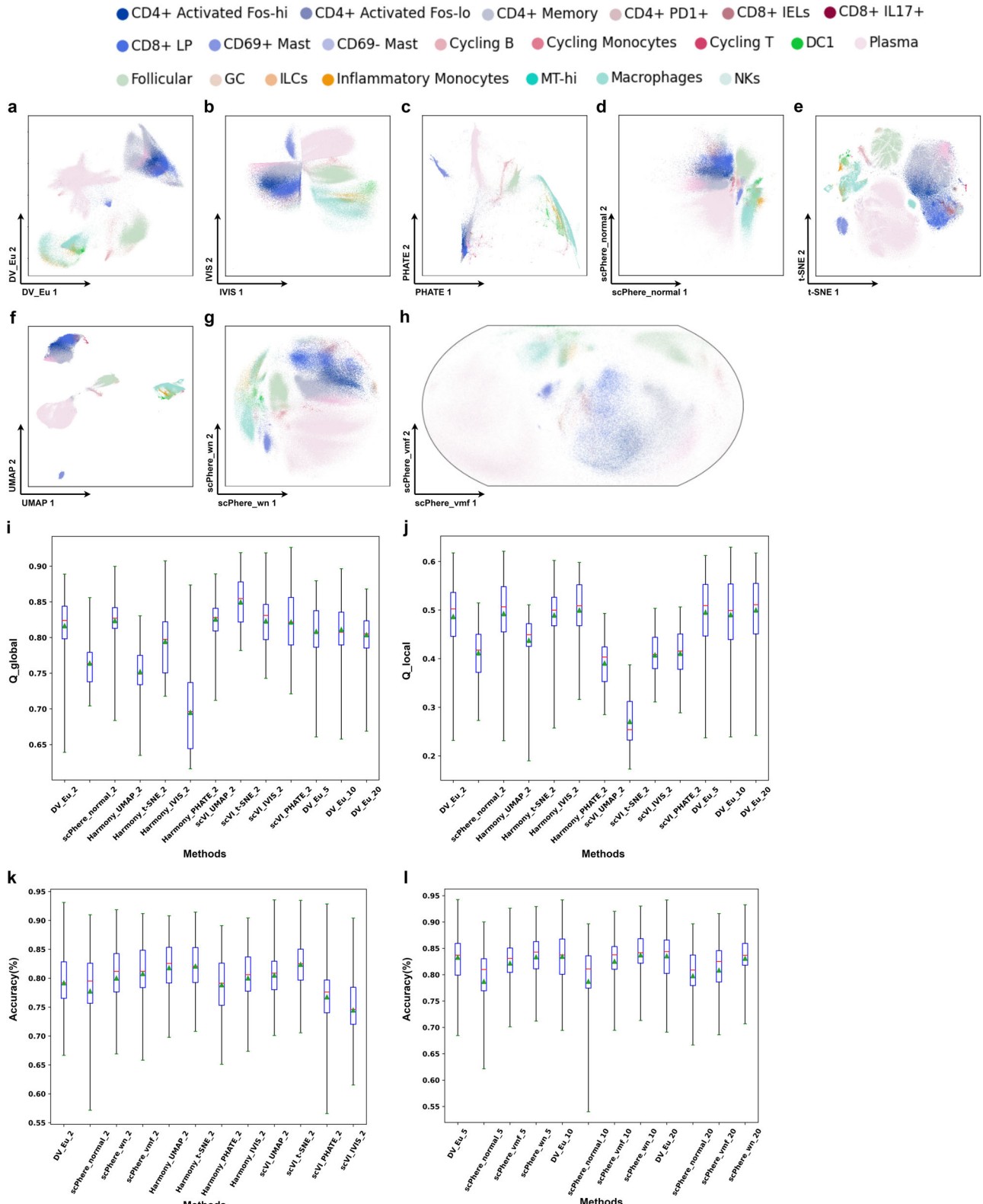

**Fig. 4 DV addresses complex technical and biological batch for visualization and analysis in colon biopsies (SCP259) from healthy individuals and UC immune patients.** 2-dimensional DV_Eu (**a**), scPhere_normal (**d**), scPhere_wn (**g**) and scPhere_vmf (**h**) embeddings accounting for the patient, location, and disease status. 2-dimensional IVIS (**b**), PHATE (**c**), t-SNE (**e**) and UMAP (**f**) embeddings (batch-corrected by Harmony accounting for the patient status). Successful batch correction visualization as reflected by local (**i**) and global (**j**) geometric structure preservation performance (y-axis), and *k*-NN classification accuracy (y axis, *k* = 5) in 2-dimensional (**k**) and low-dimensional(**l**) embeddings. The geometric structure preservation is tested on the cells from one patient between input of model and visualization embeddings. The *k*-NN classification accuracy is tested on the cells from one patient after training on the cells from all other patients. Boxplots denote means, medians and interquartile ranges (IQRs). The whiskers of a boxplot are the lowest datum still within 15 IQR of the lower quartile and the highest datum still within 15 IQR of the upper quartile.

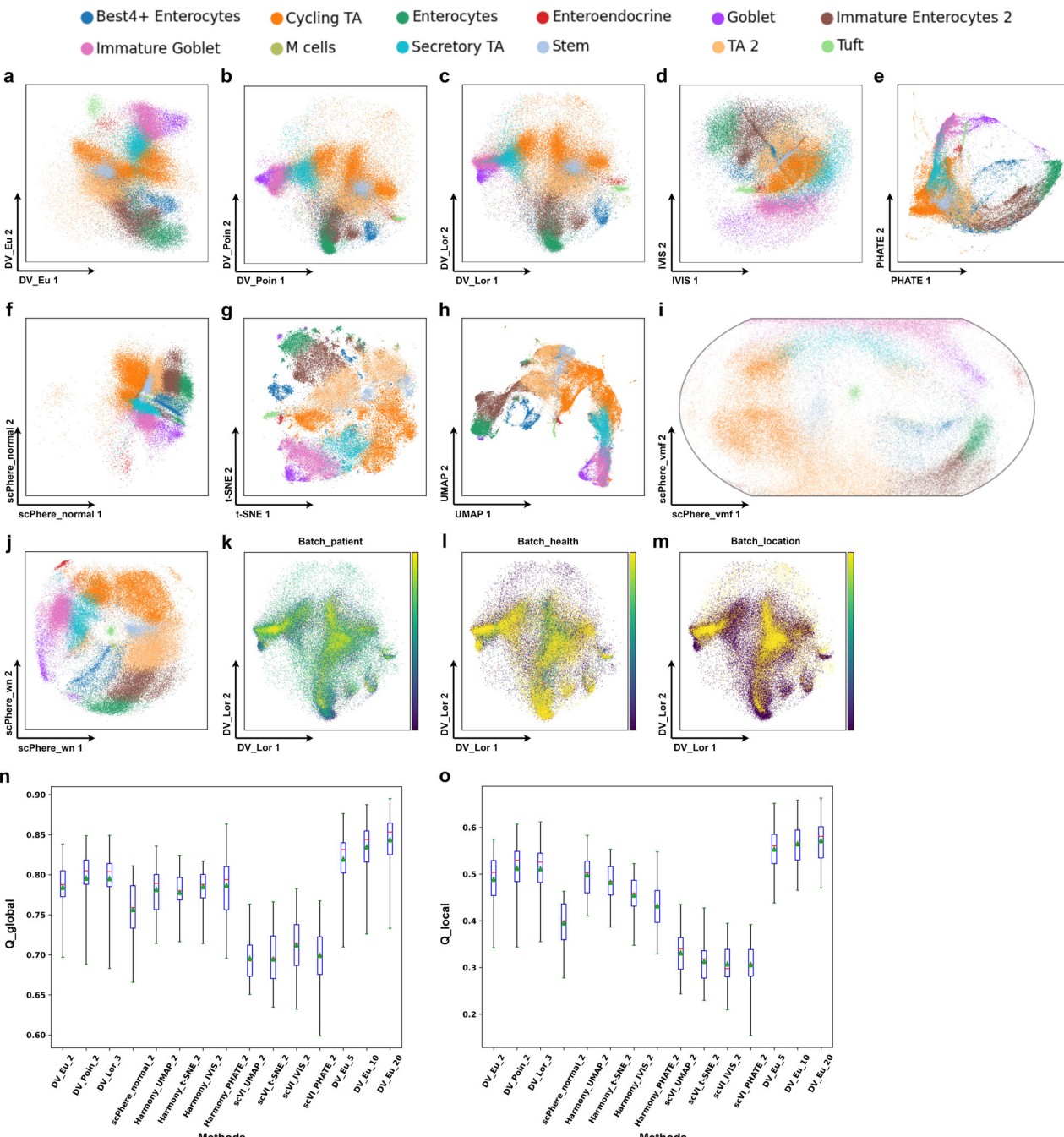

**Fig. 5 DV addresses complex technical and biological batch for visualization and analysis in colon biopsies (SCP259) from healthy individuals and UC epithelial patients.** DV_Eu (**a**), DV_Poin (**b**), DV_Lor (**c**), scPhere_normal (**f**), scPhere_wn (**j**) and scPhere_vmf (**i**) embeddings accounting for the patient, location and disease status. 2-dimensional IVIS (**d**), PHATE (**e**), t-SNE (**g**) and UMAP (**h**) embeddings (batch-corrected by Harmony accounting for the patient status). DV_Lor embeddings colored by batch_patient (**k**), batch_health (**l**) and batch_location (**m**) label. Successful batch correction visualization as reflected by local (**n**) and global (**o**) geometric structure preservation performance in 2-dimensional and low-dimensional embeddings.

while the precision of RSPO3+ is unsatisfactory. For the immune dataset (Fig. 6, Supplementary Data 4), DV_Eu (82.17%) outperforms scPhere_normal (78.23%), scPhere_wn (79.37%) and scPhere_vmf (78.58%) in terms of classification accuracy, especially on DC1 cells, DC2 cells, macrophages and cycling monocytes, while the precision decreases in inflammatory monocytes, ILC cells and DC8+IEL cells. For epithelial cells (Supplementary Fig. 8), DV and scPhere achieve similar classification accuracy, but DV possesses a better visualization ability to show two main developmental trajectories.

For heterogeneous case (training data and testing data share different gene names and numbers), we design a series of critical preprocessing methods (Fig. 1c) to overcome the heterogeneous problem, including a heterogeneous correction module (the same genes in training data and testing data are selected, for testing data, the same genes are maintained as original value, missing genes are set to 0, and redundant genes are removed), normalization, log scaling, standardization module (mean and standard deviation learned on training data is used to scale testing data), PCA module (PCA model learned on training data is used to map

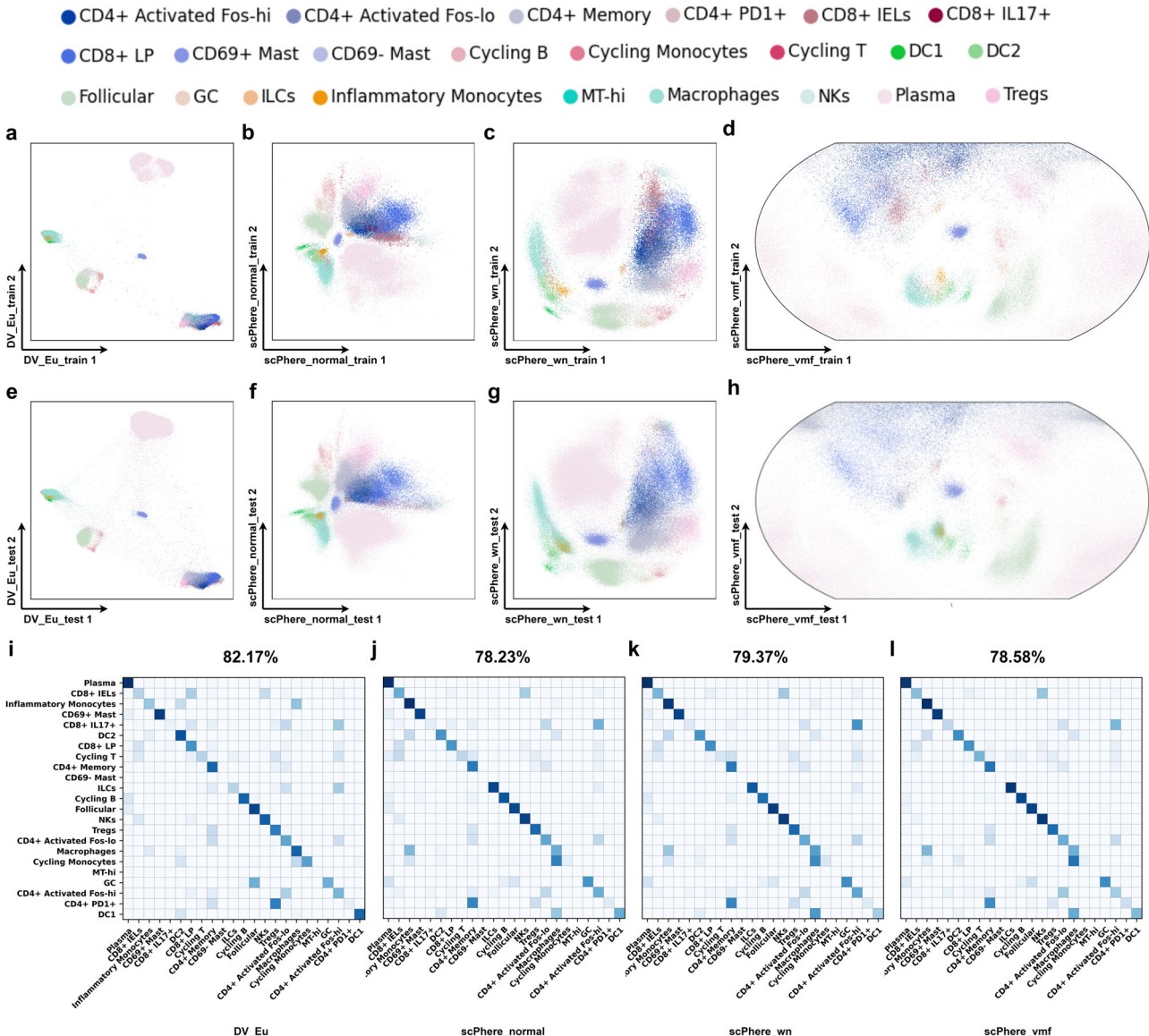

**Fig. 6 Using learned DV model to visualize cells, pinpoint cell types impacted by biological factors, and generate a "batch-invariant" reference atlas.** The "batch-invariant" DV model is trained on 18 UC immune patients (**a–d**) and tested on 12 UC immune patients (**e–h**). DV_Eu learns representations in the Euclidean space (**a, e**). 2-dimensional scPhere_normal (**b, f**), scPhere_wn (**c, g**) and scPhere_vmf (**d, h**) representations. Confusion matrices (**i–l**) of the overlap in cells (row-centered and scaled Z-score, color bar) between "true" cell types from the original study (rows) and cell assignments by k-NN classifications ($k = 5$) from "batch-invariant" DV model is trained on training set cells for immune cells (SCP259). CD69- mast cells and MThi cells are observed in the training data only.

testing data as 50 principal components). We learn a pre-trained DV_Eu model for HCL cells from 43 tissues training data, then use it to map the testing data, including the HCL cells from 28 tissues and the mouse cell atlas (MCA) cells. The experiment results demonstrate that DV's embeddings of test data with high quality even in heterogeneous cases. For example, the underlying biological information analysis in the HCL dataset mentioned above still remains, and DV_Eu obtains 72.82% classification accuracy (Supplementary Fig. 9c, only assessing the cell types presented in the training data) when data is collected in the same species and profiled by the same platform (e.g., Microwell-Seq platform). Moreover, the pre-trained DV can still locate some important clusters (e.g., Erythroid cells, Macrophage, Monocyte, B cells and Fasciculata cells, Supplementary Fig. 9d) corresponding to the training dataset when conducting cross-species experiments (e.g., training on HCL and testing on MCA).

This verifies the analysis in the original study[40] that the major cell types in mammalian organs are similar.

## Discussion

We propose the DV model to embed *static* and *dynamic* scRNA-seq cells in low-dimensional Euclidean and hyperbolic spaces to enhance exploratory data analysis and visualization of cells from single-cell studies, especially with complex multilevel batch factors. DV provides more readily interpretable representations and avoids "cell-crowding" and "cell-mixing" problems. When embedding *dynamic* cells in hyperbolic spaces, it helps to study developmental trajectories. In this case, DV_Poin and DV_Lor can place root cells near the center of a Poincaré disk automatically (distance to the center can be used as a natural definition for pseudo time). They can divide cells into multiple

regions to facilitate the analysis of cell differentiation for each major cell type individually (the cells of specified type progress continuously with distance and angle in the Poincaré disk).

The main advantage of DV is to realize the geometric structure preservation of scRNA-seq data and accounting for multilevel complex batch effects simultaneously, which disentangles cell types from patients, diseases and location variables. This is an important advantage over other DNNs methods, which fail to combine geometric structure preservation with batch correction capability. To prove it, we evaluate the effectiveness of three main components (Supplementary Fig. 17), including visualization module, structure module and batch correction module. We can harness these abilities in several ways to meet different task requirements based on the inherent characteristics of biological data: to visualize *static* or *dynamic* cells directly when dataset without batch effects, to visualize *static* or *dynamic* cells considering one factor or combination of them for dataset with multilevel complex batch effects, to investigate which cell types are most affected by a factor, or to generate general reference model, which can map new incoming homogeneous or heterogeneous data to an existing embeddings and annotate cell types. DV's ability to handle complex batch factors is an advantage over previous methods for batch correction, which handle only one batch vector. Indeed, in our benchmarking with IBD cells with 30 patients with three disease statuses, DV performs better than state-of-the-art batch correction methods such as Harmony, scVI and scPhere. In the future, we can leverage supervised information to construct a more reliable priori batch effect graph. In addition, as a parametric model, DV can naturally co-embed homogeneous or heterogeneous new incoming/testing data to a latent space learned from training data only.

DV is especially suitable for analyzing large scRNA-seq datasets: its running time scales linearly with the number of input cells (Supplementary Fig. 16). It alleviates "cell-crowding" and "cell-mixing" issues when handling with large numbers of input cells, and it can preserve local and hierarchical global geometric structures of data better than baseline methods thought its runtime slightly longer than the other methods. Finally, by learning a "batch-invariant" model that takes gene expressions or principal components as inputs to learn latent embeddings, it forms a reference to visualize and annotate new profiled cells from future studies. This is an important advantage over nonparametric methods such as t-SNE, UMAP, PHATE and Poin_maps, which do not have the ability to embed new data, especially in the presence of batch effects.

DV converges rapidly and is robust to hyperparameters. For DV models with different latent spaces (e.g., Euclidean or hyperbolic space), training is quite stable and converges rapidly. DV completes fitting with 100 epochs, and its morphology is consistent with that of 300 epochs (Supplementary Fig. 10a–j, k–t). Moreover, we adjust the hyperparameters according to the proposed typical value ranges. It can be observed that the influence of hyperparameters is limited (Supplementary Fig. 11a–f, g–m). Even if the visualization results change, it does not affect the underlying biological significance in *dynamic* scRNA-seq datasets (Supplementary Fig. 12a–g, h–n).

DV can be extended in several other ways. When cell type annotations or cell type marker genes for some of the analyzed cells are available, we can include semi-supervised learning to annotate cell types. Given the rapid development of spatial transcriptomics, single-cell ATACseq and other complementary measurements, DV can be extended for the integrative analysis of multimodal data. DV can also learn discrete hierarchical trees for better interpreting developmental trajectories using hyperbolic neural networks. Given its scope, flexibility, and extensibility, we

foresee that DV will be a valuable tool for large-scale single-cell and spatial genomics studies.

## Methods

**Data preprocessing.** The raw sequencing data is preprocessed with a series of pipelines as a common practice. The preprocessing steps consist of normalization (in the summed value), log scaling, PCA and batch correction. Among others, batch correction is crucial for complex data from multi-batches and is one of the focuses in this article. Moreover, all the compared methods but scPhere are preprocessed by default (normalization, log scaling, and PCA).

When the dimensionality of data exceeds 50 dimensions, PCA is usually applied to alleviate "the curse of dimensionality", such as in t-SNE[8], UMAP[9], IVIS[15], PHATE[20] and Poin_maps[21]. We keep top 50 principal components as the input data to all the compared methods but scPhere while the log scaling of raw data as inputs. The proposed DV method is accounting for the above two preprocessing methods.

For the traditional dimensionality reduction methods, such as PCA, t-SNE, UMAP, IVIS, PHATE and Poin_maps, require multiple separate steps (batch correction, dimensionality reduction and visualization) to achieve visualization, each with its own method or algorithm. For the batch-correction methods, such as Harmony, scVI, Seurat3 CCA and LIGER methods can only handle one batch vector, we use the patient status as the batch label. For Harmony, we use the code in "scanpy" package[43] and use the default parameter settings (e.g., dimension is set to 50). For scVI, we used the code in "scvi-tools" package[44] and use the default parameter settings (e.g., dimension is set to 10). For Seurat3 CCA and LIGER, we remove them from the experimental comparison considering their poor effect in the scPhere paper[17]. However, DV and scPhere provide an end-to-end, single process that can achieve visualization and multi-batch correction simultaneously.

We compare the proposed DV with seven previous methods, namely, PCA, t-SNE, UMAP, PHATE, Poin_maps, IVIS and scPhere. We used the code in the "scikit-learn" package[45] for PCA, t-SNE, UMAP and PHATE, and their tutorial and released code for Poin_maps, IVIS and scPhere.

**DV overview.** DV receives a scRNA-seq dataset $\mathcal{D} = \left\{ (\mathbf{x_i}, \mathbf{y_i}) \right\}_{i=1}^{n}$ as input, where $\mathbf{x_i} \in \mathbb{R}^d$ is the gene expression vector of cell $i$, $\mathbf{y_i}$ is a categorical variable vector specifying the batch label (multi-hot encoding) in which $\mathbf{x_i}$ is measured, $d$ is the number of measured genes, and $n$ is the number of cells.

For scRNA-seq data, the observed unique molecular identifier (UMI) counts of cells are sparse. Therefore, linear data augmentation (e.g., linear mixup) is adopted to improve the stability and generalization of the model, and it generates augmented data by a convex combination between cells and their $k$ neighbors based on a input graph $G_{input}$ ($k$-NN graph constructed on input data):

$$\hat{\mathbf{x}}_\mathbf{i} = (1 - r_u) \cdot \mathbf{x_i} + r_u \cdot \mathbf{x_j}, \qquad where \; \mathbf{x_j} \in \mathbf{x_i}^{N(\mathbf{x_i})}$$
$$\hat{\mathbf{y}}_\mathbf{i} = (1 - r_u) \cdot \mathbf{y_i} + r_u \cdot \mathbf{y_j}, \qquad where \; \mathbf{y_j} \in \mathbf{y_i}^{N(\mathbf{x_i})} \tag{1}$$

where $N(\mathbf{x_i})$ denotes the $k$-NN list of the data point $\mathbf{x_i}$, $r_u$ denotes the linear combination parameter and is sampled from the uniform distribution $U(0, p_u)$, $p_u$ is a hyperparameter and is set to 1. Now, we obtain the updated dataset $\widehat{\mathcal{D}} = \left\{ (\tilde{\mathbf{x}}_\mathbf{i}, \tilde{\mathbf{y}}_\mathbf{i}) \right\}_{i=1}^{a \times b} = \left\{ (\mathbf{x_i}, \mathbf{y_i}) \right\}_{i=1}^{b} \cup \left\{ (\hat{\mathbf{x}}_\mathbf{i}, \hat{\mathbf{y}}_\mathbf{i}) \right\}_{i=b+1}^{(a+1) \times b}$ combined original dataset $\mathcal{D}$ with augmented dataset $\widehat{\mathcal{D}} = \left\{ (\hat{\mathbf{x}}_\mathbf{i}, \hat{\mathbf{y}}_\mathbf{i}) \right\}_{i=1}^{a \times b}$ as the input data, where $a$ is the data augmentation number of each cell, $\hat{\mathbf{x}}_\mathbf{i}$ and $\hat{\mathbf{y}}_\mathbf{i}$ are the augmented gene expression vector of cell $i$ and the corresponding batch categorical variable vector, respectively, and here $b = n$ due to DV is trained by mini-batch stochastic gradient descent.

**Visualization module.** Although $\mathbf{x_i}$ is high-dimensional, its intrinsic dimensionality is typically much lower. Manifold learning assumes that a decent embedding should preserve the geometric structure of data as much as possible. Therefore, we optimize DV based on the geometric structure preservation loss function, which minimizes the distribution discrepancy between $G_{structure}$ and $G_{visualization}$ in the form of fuzzy sets cross entropy (two-way divergency):

$$L_{GSP} = \sum_{i,j=1}^{(a+1) \times b} \mathbf{u}_{ij}^{\mathbf{st}} \log \frac{\mathbf{u}_{ij}^{\mathbf{st}}}{\mathbf{u}_{ij}^{\mathbf{vi}}} + \left(1 - \mathbf{u}_{ij}^{\mathbf{vi}}\right) \log \frac{\left(1 - \mathbf{u}_{ij}^{\mathbf{vi}}\right)}{\left(1 - \mathbf{u}_{ij}^{\mathbf{st}}\right)} \tag{2}$$

where $b$ is the number of batch size, $\mathbf{u}_{ij}^{\mathbf{st}}$ is the undirectional similarity between structure embedding $\mathbf{z}_\mathbf{i}^{\mathbf{st}}$ and $\mathbf{z}_\mathbf{j}^{\mathbf{st}}$ learned by structure module, and $\mathbf{u}_{ij}^{\mathbf{vi}}$ is the undirectional similarity between visualization embedding $\mathbf{z}_\mathbf{i}^{\mathbf{vi}}$ and $\mathbf{z}_\mathbf{j}^{\mathbf{vi}}$ learned by visualization module. The undirectional similarity $\mathbf{u}_{ij}$ is defined as following:

$$\mathbf{u}_{ij} = \mathbf{u}_{i|j} + \mathbf{u}_{j|i} - 2\mathbf{u}_{i|j}\mathbf{u}_{j|i} \tag{3}$$

where $\mathbf{u}_{j|i}$ is a directional similarity (edge weigt of graph) converted from the Euclidean or hyperbolic distance $D(\tilde{\mathbf{z}}_\mathbf{i}, \tilde{\mathbf{z}}_\mathbf{j})$ between embedding $\tilde{\mathbf{z}}_\mathbf{i}$ and embedding $\tilde{\mathbf{z}}_\mathbf{j}$

and adopts normalized squared $t$-distribution:

$$\mathbf{u}_{j|i}(\nu) = g(D(\tilde{\mathbf{z}}_i, \tilde{\mathbf{z}}_j)|\nu) = C_\nu \left(1 + \frac{D(\tilde{\mathbf{z}}_i, \tilde{\mathbf{z}}_j)}{\nu}\right)^{-(\nu+1)} \quad (4)$$

where $\nu$ is the degrees of freedom in the t-distribution, $\nu^{st}$ in $\mathbf{u}_{j|i}^{st}$ is set to 100, $\nu^{vi}$ in $\mathbf{u}_{j|i}^{vi}$ is a hyperparameter, and

$$C_\nu = 2\pi \left(\frac{\Gamma\left(\frac{\nu+1}{2}\right)}{\sqrt{\nu}\pi\Gamma\left(\frac{\nu}{2}\right)}\right)^2 \quad (5)$$

is the normalizing function of $\nu$.

**Structure module**. Especially, given that the observed UMI counts of cells are sparse, the relationship between cells is difficult to be defined by vector similarity (e.g., Euclidean distance) directly. Therefore, to estimate the local geometries of the underlying topological manifold and construct a more reliable $G_{structure}$ to describe the relationship between cells, we make a local scale contraction for Euclidean distance between structure embedding $\mathbf{z}_i^{st}$ of each cell and structure embedding $\hat{\mathbf{z}}_i^{st}$ of its corresponding augmented cells for $G_{structure}$, and the corresponding $\mathbf{u}_{j|i}^{st}$ is redefined as:

$$\mathbf{u}_{j|i}^{st}(\nu) = g\left(D(\mathbf{z}_i^{st}, \hat{\mathbf{z}}_i^{st})|\nu\right) = C_\nu \left(1 + \frac{D_\mathbb{E}(\mathbf{z}_i^{st}, \hat{\mathbf{z}}_i^{st})/\gamma}{\nu}\right)^{-(\nu+1)} \quad (6)$$

where $\gamma$ is the local scale contraction coefficient.

**Batch correction module**. In addition, to remove batch effect problem simultaneously, we introduce a priori batch effect graph $G_{batch}$ constructed on batch categorical variable vector based on Euclidean distance $D_\mathbb{E}(\tilde{\mathbf{y}}_i, \tilde{\mathbf{y}}_j)$ merged with $G_{visualization}$ in the training process (Fig. 1b). Therefore, the $\mathbf{u}_{j|i}^{vi}$ is redefined as:

$$\mathbf{u}_{j|i}^{vi}(\nu) = g(D(\tilde{\mathbf{z}}_i^{vi}, \tilde{\mathbf{z}}_j^{vi})|\nu) = C_\nu \left(1 + \frac{D(\tilde{\mathbf{z}}_i^{vi}, \tilde{\mathbf{z}}_j^{vi}) + \beta \cdot D_\mathbb{E}(\tilde{\mathbf{y}}_i, \tilde{\mathbf{y}}_j)}{\nu}\right)^{-(\nu+1)} \quad (7)$$

where $\beta$ represents the importance of $G_{batch}$.

**Poincaré ball and Lorentz model of the hyperbolic space**. A Riemannian manifold $(\mathbb{M}, g)$ is a real and smooth manifold equipped with an inner product $g_\mathbf{x} : \mathbb{T}_\mathbf{x}\mathbb{M} \times \mathbb{T}_\mathbf{x}\mathbb{M} \to \mathbb{R}$ at each point $\mathbf{x} \in \mathbb{M}$, which is called a Riemannian metric and allows us to define the geometric properties of a space such as angles and the length of a curve. We introduce two commonly used hyperbolic manifolds compared with the Euclidean manifold:

The Euclidean manifold is a manifold with zero curvature. The metric tensor is defined as $g^\mathbb{E} = \text{diag}([1, 1, \ldots, 1])$. The closed-form distance, i.e., the length of the geodesic, which is a straight line in Euclidean space, between two points is given as:

$$D_\mathbb{E}(\mathbf{x}_i, \mathbf{x}_j) = \sqrt{(\mathbf{x}_i - \mathbf{x}_j)^T(\mathbf{x}_i - \mathbf{x}_j)} \quad (8)$$

The exponential map of the Euclidean manifold is defined as:

$$\exp_\mathbf{x}(\mathbf{v}) = \mathbf{x} + \mathbf{v} \quad (9)$$

The Poincaré ball model with constant negative curvature $-K(K > 0)$ corresponding to the Riemannian manifold $(\mathbb{P}, g_\mathbf{x}^\mathbb{P})$, where $\mathbb{P} = \left\{\mathbf{x} \in \mathbb{R}^d : \|\mathbf{x}\| < \frac{1}{K}\right\}$ is an open ball. The metric tensor is defined as $g_\mathbf{x}^\mathbb{P} = (\lambda_\mathbf{x}^K)^2 g^\mathbb{E}$, where $\lambda_\mathbf{x}^K = \frac{2}{1+K\|\mathbf{x}\|^2}$ is the conformal factor and $g^E$ is the Euclidean metric tensor. The origin of $\mathbb{P}$ is $\mathbf{o} = (0, \ldots, 0) \in \mathbb{R}^d$. The distance between two points $\mathbf{x}_i, \mathbf{x}_j \in \mathbb{P}$ is given as:

$$D_\mathbb{P}(\mathbf{x}_i, \mathbf{x}_j) = \frac{1}{\sqrt{K}} \text{arcosh}\left(1 + 2K\frac{\|\mathbf{x}_i - \mathbf{x}_j\|^2}{(1 - K\|\mathbf{x}_i\|^2)(1 - K\|\mathbf{x}_j\|^2)}\right) \quad (10)$$

For any point $\mathbf{x}_i \in \mathbb{P}$, the exponential map $\exp_\mathbf{x} : \mathbb{T}_\mathbf{x}\mathbb{P} \to \mathbb{P}$ is defined for the tangent vector $\mathbf{v} \neq 0$ and the point $\mathbf{x}_j \neq 0$ as:

$$\exp_\mathbf{x}^K(\mathbf{v}) = \mathbf{x} \oplus_K \left(\tanh\left(\frac{\sqrt{K}\lambda_\mathbf{x}^K \|\mathbf{v}\|}{2}\right)\frac{\mathbf{v}}{\sqrt{K} \|\mathbf{v}\|}\right) \quad (11)$$

where $\oplus_K$ is the **Möbius addition** for any $\mathbf{x}_i, \mathbf{x}_j \in \mathbb{P}$:

$$\mathbf{x}_i \oplus_K \mathbf{x}_j = \frac{(1 + 2K\langle\mathbf{x}_i, \mathbf{x}_j\rangle - K\|\mathbf{x}_j\|^2)\mathbf{x}_i + (1 - K\|\mathbf{x}_i\|^2)\mathbf{x}_j}{1 + 2K\langle\mathbf{x}_i, \mathbf{x}_j\rangle + K^2\|\mathbf{x}_i\|^2\|\mathbf{x}_j\|^2} \quad (12)$$

The Lorentz model avoids numerical instabilities that may arise with the Poincaré distance (mostly due to the division). Let $\mathbf{x}_i, \mathbf{x}_j \in \mathbb{R}^{d+1}$, then the Lorentzian scalar product is defined as:

$$\langle\mathbf{x}_i, \mathbf{x}_j\rangle_\mathbb{L} = -(\mathbf{x}_i)_0(\mathbf{x}_j)_0 + \sum_{i=1}^d (\mathbf{x}_i)_d(\mathbf{x}_j)_d \quad (13)$$

The Lorentz model with constant negative curvature $-K(K > 0)$ corresponding to the Riemannian manifold $(\mathbb{L}, g_\mathbf{x}^\mathbb{L})$, where $\mathbb{L} = \left\{\mathbf{x} \in \mathbb{R}^{d+1} : \langle\mathbf{x}, \mathbf{x}\rangle_\mathbb{L} = -1, \mathbf{x}_0 > 0\right\}$ and where $g^\mathbb{L} = \text{diag}([-1, 1, \ldots, 1])$. The induced distance function is given as:

$$D_\mathbb{L}(\mathbf{x}_i, \mathbf{x}_j) = \sqrt{K}\text{arcosh}(-\langle\mathbf{x}_i, \mathbf{x}_j\rangle_\mathbb{L}/K) \quad (14)$$

The exponential map $\exp_\mathbf{x} : \mathbb{T}_\mathbf{x}\mathbb{L} \to \mathbb{L}$ is defined as:

$$\exp_\mathbf{x}^K(\mathbf{v}) = \cosh\left(\frac{\|\mathbf{v}\|_\mathbb{L}}{\sqrt{K}}\right)\mathbf{x} + \sqrt{K}\sinh\left(\frac{\|\mathbf{v}\|_\mathbb{L}}{\sqrt{K}}\right)\frac{\mathbf{v}}{\|\mathbf{v}\|_\mathbb{L}} \quad (15)$$

where $\|\mathbf{v}\|_\mathbb{L} = \sqrt{\langle\mathbf{v}, \mathbf{v}\rangle_\mathbb{L}}$. The origin, i.e., the zero vector in Euclidean space and the Poincaré ball, is equivalent to $(1, 0, \ldots, 0)$ in the Lorentz model. The point $(\mathbf{x}_0, \mathbf{x}_1, \ldots, \mathbf{x}_n)^T$ between Poincaré ball and Lorentz model can be conveniently converted:

$$p_{\mathbb{L}\to\mathbb{P}}(\mathbf{x}_0, \mathbf{x}_1, \ldots, \mathbf{x}_n) = \frac{(\mathbf{x}_1, \ldots, \mathbf{x}_n)}{\sqrt{K}\mathbf{x}_0 + 1} \quad (16)$$

**Model structure**. We use the Leaky rectified linear units (LeakyReLU) activation functions for hidden layers. The gradient can also be calculated where the input of LeakyReLU activation function is less than zero during the backpropagation process, avoiding the jagged problem in the direction of the gradient. Meanwhile, we use batch normalization (BN) for hidden layers. In the training process of DNNs, the input of each layer can keep the same distribution.

For all experiments, we use a six-layered neural network, including a structure module (d/50-500-300-100) and a visualization module (100-300-100-3/2). The dimensionality of the visualization embedding layer was typically 2/3 for visualization purposes. When comparing DV based on different dimensions of latent spaces (e.g., 2/3, 5, 10 and 20), we keep all other factors the same. We use the Adam stochastic optimization algorithm and train model for 300 epochs. For our current implementation, we do not introduce an early stopping but train DV for a given number of epochs. We can obtain a good embedding when we only train the model for 50 epochs. We run all the experiments using a Ubuntu server and a single V100 GPU with 32GB memory.

**Choices of hyperparameters**. To ensure that each method achieves its optimal performance, we use the grid search method to find the optimal hyperparameter. For t-SNE, the search space of "perplexity" is {5, 10, 15, 20, 30, 40}. For UMAP, the search space of "min_dist" is {0.1, 0.3, 0.5, 0.7, 0.9}. For IVIS, the search space of "k" is {2, 4, 8, 16, 32, 64}. For Poin_maps, the search space of "knn" is {15, 20, 30}, "sigma" is {1, 2} and "gamma" is {1, 2}. For PHATE, we use the default hyper-parameter settings. For scPhere, we use the default hyperparameter setting according to the released code.

In the following, we discuss the function of different hyperparameters in DV and propose typical value ranges. The "learning rate" adjusts the objective function converge to the local minimum in the proper time and the search space is typically set to $\{1e^{-3}, 5e^{-3}\}$. The "batch size" improves the memory utilization and the accuracy of gradient descent direction, and the search space is typically set to {500, 1000, 2000}. The "$\nu^{vi}$" controls the magnitude of edge weight in $G_{visualization}$ and the search space is typically set to $\{1e^{-3}, 5e^{-3}, 1e^{-2}\}$. The "$\gamma$" controls the local scale contraction level between each cell and its corresponding augmented cells and the search space is typically set to $\{10, 1000, 1e^5\}$. The "$\beta$" controls how much batch information need to remove in loss function and the search space is typically set to $\{1e^{-2}, 1, 100\}$. The hyperparameters used by DV_Eu, DV_Poin and DV_Lor for each experimental dataset are shown in Supplementary Tabs. 1-3, respectively.

**Quantifying global/hierarchical and local structure preservation**. To quantitatively compare the performance of different visualization methods, we use the scale-independent quality criteria proposed by Lee and Verleysen[46] following Poin_maps[21]. The main idea of this method is that a good dimensionality reduction method should have good preservation of local and global distances on the manifold, e.g., close neighbors should be placed close to each other while maintaining large distances between distant points. Therefore, they proposed to use two scalar quality criteria $Q_{local}$ and $Q_{global}$ focusing separately on low- and high-dimensional qualities of the embeddings. The quantities of $Q_{local}$ and $Q_{global}$ range from 0 (bad) to 1 (good) and reflect how well are local and global properties of the dataset are preserved in the embeddings. To estimate distances in the high-dimensional space, we use Euclidean distances estimated as the length in a full-connected graph. For Poin_maps, we follow its released code and use the geodesic distances estimated as the length of a shortest-path in a $k$-nearest neighbors graph. For distances in the low-dimensional space, we use Euclidean distances except DV_Poin, DV_Lor, and Poin_maps methods, for which we use hyperbolic distances. Furthermore, for datasets with batch effect problem, we evaluate the geometric structure preservation performance of each batch separately, calculate the $Q_{local}$ and $Q_{global}$ between the input data without batch correction and output embeddings, and use the boxplots for statistics.

To learn a DV model that is invariant to the batch vectors and can be used to map cells from completely new batches, we use DV model to map a gene expression vector to the low-dimensional representation directly without using the batch vector as an input to the model when training the DV model. The batch vector is only used in the objective function that takes both the latent representation of a cell and its cell batch vector to construct the latent geometric structure to preserve the semantic geometric structure during training DV. We call this modality of DV with no batch vectors for the "batch-invariant" DV, as it learns latent representations that are invariant to the batch vectors.

**Statistics and reproducibility**. The details about experimental design and statistics used in different data analyses performed in this study are given in the respective sections of results and methods.

**Reporting summary**. Further information on research design is available in the Nature Portfolio Reporting Summary linked to this article.

## Data availability

We use publicly available datasets in this study (GEO: GSE126954[35], GSE119562[38], GSE130148[36], GSE111588[37], GSE137400[39], GSE134355[40], GSE126954[41]; Single Cell Portal: SCP259, SCP551. To make the results presented in this study reproducible, all processed data are available in Single Cell Portal SCP1873.

## Code availability

The DV software package, implemented in Pyotrch, is available free from https://github.com/Westlake-AI/DV, and as a Supplementary Software 1 accompanying this manuscript.

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

## Acknowledgements

This work is supported in part by the Science and Technology Innovation 2030 - Major Project (No. 2021ZD0150100) and National Natural Science Foundation of China (No. U21A20427).

## Author contributions

S.Z.L. proposed this research, Y.X. Z.Z., and S.Z.L. developed the method, Y.X., J.X., C.T., and Y.G. collected the datasets, Y.X. conceived the experiment and wrote the manuscript with guidance from S.Z.L., and Y.X. conducted the experiment and analyzed the results. All authors discussed the results, revised the draft manuscript, and read and approved the final manuscript.

## Competing interests

The authors declare no competing interests.
