## [Peer Review File · Communications Biology]

Reviewers' comments:

Reviewer #1 (Remarks to the Author):

Xu and coauthors introduced a data visualization method for high-dimensional scRNA-seq data. Empirical results show very promising results on several large datasets. Here I have several major comments, most on the methods for the authors to address.

1, it's still unclear whether the inputs to DV are raw counts or PCs. It seems to be PC but was not written in any places. However, on pg. 16, the authors said k-NN graph was constructed on raw data. It's confusion about the 'raw data' being the raw UMI counts or PCs. If they are counts, how do the authors calculate the distances between two cells?

2, how are the similarities in high-dimensional data defined, based on Euclidean distances of PCs?

3, how to set the pu parameter (pg. 16) is not given?

4, The authors did a grid search for parameters (p18), and I'm not sure the final parameters for each dataset, could be helpful to present such information in a table.

5, it's not clear about the computational efficiency of the DV, since the objective function involves a normalized t-distribution, is it time-consuming compared to other methods such as scsphere?

6, the DV algorithm has several components (steps), e.g., data argumentations and cross-entropy loss. Which component contributes most to the success of the algorithm?

7, why only scsphere_normal is presented for comparison in Fig. 5n-o?

8, From Fig.3-6, it seems that DV_Euc typically works pretty well. If that's the case, is there any need to use non-Euclidean embedding?

9, It seems that the data and software tools are not available for reviewers (and users)?

Minors:

1, Fig. 4,5 boxplots x-axis text misaligned.

2, Fig. 6 colors don't align in training and test data?

Reviewer #2 (Remarks to the Author):

In this manuscript, Xu et al. reported a new dimension reduction method DV for single cell RNA-seq data visualization. The authors claimed DV can preserve inherent structure of data from different batches. The topics and problems discussed in this manuscript have a broad audience as the low dimension representation is one of the most forward ways to interpret big single-cell data. The authors also used sufficient data to do benchmarks. This is a promising work, but there are several aspects that may need to be improved:

1. It is very hard to get the dataset information in each figure. As the authors used a lot of datasets in their manuscript, they should list the related dataset ID(GSE***) in each figure legend. I point this because the integrity of some statements I need to check from original paper of the related datasets. For example, in Line 134-135, the authors state that in adipose tissue stromal cells, all methods mixed pericyte and APC except DV_Lor. The annotation information of this dataset should be from the

original paper, I am curious to know if the APC and pericyte separated in original umap or tsne.

2. It is not very clear the differences between semantic module and visualization module. The only difference I can get from the paper is the number of dimension.

3. Did authors use all features or variable features when ran umap, tsne and other method?

4. Because others methods can not handle the batch effects, it is no surprise that the performance of DV is better than performances of other methods. I think the authors should also compare the DV vs (batch effect remove method, like harmony, scvi)+(other dimension reduction methods) as this is the most common pipeline for analyzing multiple batch datasets.

Dear reviewers,

We are grateful for the valuable comments and suggestions of the reviewers. We make several major revisions in accordance with the suggestions, including but not limited to the following:

- (1) A comparison is made with standard procedures to do batch correction and visualization such as using a batch correcting tool followed by applying an independent dimensionality reduction method. The compared methods are the following:
 - a) Harmony+t-SNE, Harmony+UMAP, Harmony+ IVIS, Harmony+ PHATE;
 - b) scVI + t-SNE, scVI + UMAP, scVI + IVIS, scVI + PHATE;
- (2) Specifications are added about the hyperparameters used by DV for each experimental dataset based on the reviewers' comments.
- (3) More ablation study experiments are added to evaluate the importance of data augmentation, batch graph and cross-entropy loss in DV.
- (4) Information about the runtime and computational load of all algorithms is provided.
- (5) The overall presentation is improved, including the following:
 - a) More appropriate terms are used, replacing “semantic” with “structure” module/graph/embedding, replacing “discrete” with “*static* (at a time point)”, replacing “continuous” with “*dynamic* (at a sequence of time points)”, and replacing “raw data” with “input data”.
 - b) The description of the DV framework in Figure 1 is improved.
 - c) Errors and typos are corrected.
- (6) The DV code and data will be released upon acceptance.

embedding reflecting high-dimensional data information to construct more reliable structure graph (replacing semantic graph). Therefore, the similarities in high-dimensional data are defined using student t-distribution based on Euclidean distances of structure embedding.

3. How to set the p_u parameter (pg. 16) is not given?

A: Thank you so much for this question. In the revised manuscript, we add this information. The p_u parameter is set to 1 by default (in Line 454) in the `torch.rand(·)` function, which means uniformly random sampling between 0 and 1.

4. The authors did a grid search for parameters (p18), and I'm not sure the final parameters for each dataset, could be helpful to present such information in a table.

A: Thank you so much for this instructive comment. In the revised manuscript, we add the hyperparameters used by DV for each experimental dataset as shown in Table 1, Table 2 and Table 3 of Supplementary Information. Moreover, we also discuss the robustness (in Line 343~346) of DV to hyperparameter on processing large *static* (“discrete” in the previous version) and *dynamic* (“continuous”) scRNA-seq data with batch effect problem (Supplementary Fig. 11 and Fig. 12).

5. It's not clear about the computational efficiency of the DV, since the objective function involves a normalized t-distribution, is it time-consuming compared to other methods such as scSphere?

A: Thank you so much for this suggestion. In the revised manuscript, we add the runtime to empirically check the time complexity and computational load of all algorithms as shown in Fig. 16 of Supplementary Information. It alleviates “cell-crowding” and “cell-mixing” issues when handling with large numbers of input cells, and it can preserve local and hierarchical global geometric structures in data better than baseline methods though its runtime is slightly longer than the other methods. Theoretically, the vMF distribution and wrapped normal distribution used in scSphere also increase the running time of the algorithm. Therefore, the normalized t-distribution in objective function is not the key to the time consuming of the algorithm compared to scSphere. In the future version of code, we will further improve the efficiency of the code, such as saving the input graph structure constructed on

input data in advance and improving the efficiency of code calling GPU.

[1] Ding J, Regev A. Deep generative model embedding of single-cell RNA-Seq profiles on hyperspheres and hyperbolic spaces[J]. Nature communications, 2021, 12(1): 1-17.

6. The DV algorithm has several components (steps), e.g., data augmentations and cross-entropy loss. Which component contributes most to the success of the algorithm?

A: Thank you so much for this instructive comment. In the revised manuscript, we add the ablation study experiment to compare the importance of data augmentation, batch graph and cross-entropy loss as shown in Fig. 17 of Supplementary Information. It can be noticed that data augmentation contributes most to the success of DV. In addition, we replace the cross-entropy loss with L_2 norm, the degree of separation between clusters decreased significantly in UC stromal and UC epithelial datasets, but this phenomenon is not obvious in UC immune dataset. When we remove the batch graph module, DV lose the batch correction ability on UC epithelial dataset, the degree of separation between clusters decreased significantly in UC stromal and UC immune datasets, which implies that the structure module cannot learn reliable structure embeddings for datasets with batch effect problem.

7. Why only scSphere_normal is presented for comparison in Fig. 5n-o?

A: Thank you so much for this question. Here we want to make some clarifications. scSphere based on variational autoencoders and does not require a distance metric. Therefore, we do not find spherical and hyperbolic distances in the released scSphere code that can accurately match the spherical and hyperbolic embedding of scSphere. If we define spherical and hyperbolic distances ourselves (which we actually do), we find that the geometric structure preservation index Q_{local} and Q_{global} of scSphere_vmf and scSphere_wn are poor, which we believe is unfair. Therefore, we only compare scSphere_normal, which can use Euclidean distance directly. In the revised manuscript, we add more baseline methods in Fig.5 n-o which are sufficient to demonstrate the effectiveness of DV.

8. From Fig.3-6, it seems that DV_Eu typically works pretty well. If that's the case, is

*lot of datasets in their manuscript, they should list the related dataset ID(GSE***) in each figure legend. I point this because the integrity of some statements I need to check from original paper of the related datasets. For example, in Line 134-135, the authors state that in adipose tissue stromal cells, all methods mixed pericyte and APC except DV_Lor. The annotation information of this dataset should be from the original paper, I am curious to know if the APC and pericyte separated in original umap or tsne.*

A: Thank you so much for this instructive comment. In the revised manuscript, we list the related dataset ID (GSE***) in each figure legend and modify the “adipose tissue stromal cells” (writing error) to “human lung cells”. In terms of the human lung cells, we adopt the preprocessed data in scPHERE. We look at the results of t-SNE and UMAP in scPHERE's paper (Supplementary Fig. 2 of scPHERE) and find that the results (Supplementary Fig. 1 q, r of this manuscript) are basically consistent with them.

[1] Ding J, Regev A. Deep generative model embedding of single-cell RNA-Seq profiles on hyperspheres and hyperbolic spaces[J]. Nature communications, 2021, 12(1): 1-17.

2. It is not very clear the differences between semantic module and visualization module. The only difference I can get from the paper is the number of dimension.

A: Thank you so much for this question. In the revised manuscript, we replace “semantic module” with “structure module” to be more precise, and add titles of visualization module, structure module and batch correction module in “DV overview” section on pg.16. For the structure module, the goal is to estimate the local geometry of the underlying topological manifold, and learn structure embedding reflecting high-dimensional data information to construct more reliable structure graph through making a local scale contraction for Euclidean distance between structure embedding of each cell and structure embedding of its corresponding augmented cells (Equation (6) on pg. 17). For the visualization module, the goal is to preserve the geometry structure of structure embedding as much as possible based on geometric structure preservation loss function (Equation (2) on pg. 16).

3. Did authors use all features or variable features when ran umap, tsne and other

method?

A: Thank you so much for this question. As described in the “data preprocessing” section on pg. 16, DV, t-SNE, UMAP, IVIS, PHATE and Poincaré maps use 50 PCs (normalization, log scaling, and PCA) as inputs by default (in Line 431). Moreover, we compare DV using the raw data only preprocessed by log scaling as inputs with t-SNE, UMAP and scPhere (in Line 269~275, Supplementary Fig. 6). For scPhere, we use the raw data only preprocessed by log scaling as inputs according to the released code.

4. Because others methods can not handle the batch effects, it is no surprise that the performance of DV is better than performances of other methods. I think the authors should also compare the DV vs (batch effect remove method, like harmony, scvi)+(other dimension reduction methods) as this is the most common pipeline for analyzing multiple batch datasets.

A: We agree with and accept this suggestion. In fact, the results of baseline methods (Fig. 4, Fig. 5 and Supplementary Fig. 4) in original manuscript are (Harmony + other dimension reduction methods). In the revised manuscript, we thoroughly compare the effectiveness of DV with (batch effect remove method, like Harmony, scVI) + (other dimension reduction methods, like t-SNE, UMAP, IVIS, PHATE) in Fig.4, Fig.5 and Supplementary Fig.4. It can be noticed that DV still shows good results (in Line 234~245).

REVIEWERS' COMMENTS:

Reviewer #1 (Remarks to the Author):

The authors addressed all my comments.